# Stereoselective oxidative glycosylation of anomeric nucleophiles with alcohols and carboxylic acids

Tianyi Yang[1], Feng Zhu[1] & Maciej A. Walczak [1]

Oligosaccharides, one of the most abundant biopolymers, are involved in numerous biological processes. Although many efforts have been put in preparative carbohydrate chemistry, achieving optimal anomeric and regioselectivities remains challenging. Herein we describe an oxidative glycosylation method between anomeric stannanes and oxygen nucleophiles resulting in the formation of a C−O bond with consistently high anomeric control for glycosyl donors bearing a free C2-hydroxyl group. These reactions are promoted by hypervalent iodine reagents with catalytic or stoichiometric amounts of Cu or Zn salts. The generality of this transformation is demonstrated in 42 examples. Mechanistic studies indicate that the oxidative glycosylation is initiated by the hydroxyl-guided delivery of the hypervalent iodine and tosylate into the anomeric position, and results in excellent 1,2-*trans* selectivity. The unique mechanistic paradigm, high selectivities, and mild reaction conditions make this method suitable for the synthesis of oligosaccharides and for integration with other methodologies such as automated synthesis.

[1] Department of Chemistry and Biochemistry, University of Colorado Boulder, Boulder, CO 80309, USA. These authors contributed equally: Tianyi Yang, Feng Zhu.  Correspondence and requests for materials should be addressed to M.A.W. (email: maciej.walczak@colorado.edu)

Saccharides are essential biomolecules that, due to their vast chemical diversity, fulfill a wide scope of physiological functions ranging from the maintenance and survival of cells to the storage and supply of energy[1]. The inherent structural complexity of carbohydrates enables them to interact with myriad biological receptors and the efforts to decipher the sugar code has recently received considerable attention[2]. One strategy to understand the role of saccharides in living organisms is to prepare well-defined glycans that, when used as probes, provide a meaningful output that can be correlated with the structure of the sugar. An approach that capitalizes on chemical synthesis is characterized by broad versatility and scalability as any desired modifications are available by chemical means. However, a great majority of methods that form the glycosidic bond often relies on displacement reactions at the C1 position making the control of anomeric configuration a daunting task[3]. The introduction of protective groups is often deemed necessary in order to regioselectively install a new C−O linkage between two saccharide groups, despite obvious limitations such as suboptimal step-economy and synthetic efficiency. Without enzymatic catalysis, direct glycosylations engaging glycosyl donors bearing free hydroxyl groups are known for their low yields and poor selectivities[4,5]. Furthermore, stereoselective chemical glycosylations that are characterized by high anomeric selectivities across a broad range of substrates often rely on neighboring group participation, transient protection, or the unique scaffold features of the glycosyl donor (e.g., 1,2-anhydrosugars also known as glycal epoxides)[6]. A conceptually orthogonal approach that addresses the problem of variable anomeric selectivities requires the use of glycosyl donors with free hydroxyl groups that can act as a directing group thus eliminating the need for exhaustive protecting group manipulations. Herein, we report one-pot oxidative O-glycosylation of partially protected glycosyl donors that proceeds under mild reaction conditions without compromising yield and with exclusive anomeric selectivities.

Commonly employed strategies for stereoselective O-glycosylation are outlined in Fig. 1. Of these, the most widely applied methods capitalize on reactions of glycosyl donors in which the C2 position is capped with a carbonyl-based participating group (e.g., OAc, OBz, NPhth)[7–10]. For example, activation of 2-O-acyl thioglycoside with an oxidant and triflate salt leads to the formation of the dioxolenium ion intermediate 4 which is quenched with an alcohol to provide glycoside 1 in high 1,2-trans selectivity (Fig. 1a)[11]. Other approaches based on a locked bicyclic conformation utilize acetal[12], carbonate[13], or carbamate[14] groups introduced into the pyranose scaffold to amplify the differences of the energy barriers between two transition states[15]. Ring opening reactions of glycal epoxides in the presence of Lewis acid catalysts or bases provided a more direct solution to 1,2-trans configured glycosides (Fig. 1b)[16–19]. Primary alcohols and sterically unhindered secondary alcohols found the most success with this method, while some demanding glycosyl acceptors depend more on the pre-activation of hydroxyl group (e.g., as a stannyl ether) and the compatible Lewis acid to achieve good yield and stereoselectivity[17,20,21].

The problem of anomeric selectivities is even more pronounced when encountered in reactions with partially protected or free saccharides. In addition to the control of the anomeric configuration, regioselectivity becomes a critical consideration. Unlike reactions of partially protected glycosyl acceptors that can be manipulated using catalytic or stoichiometric conditions, only isolated examples of chemical glycosylations have been reported

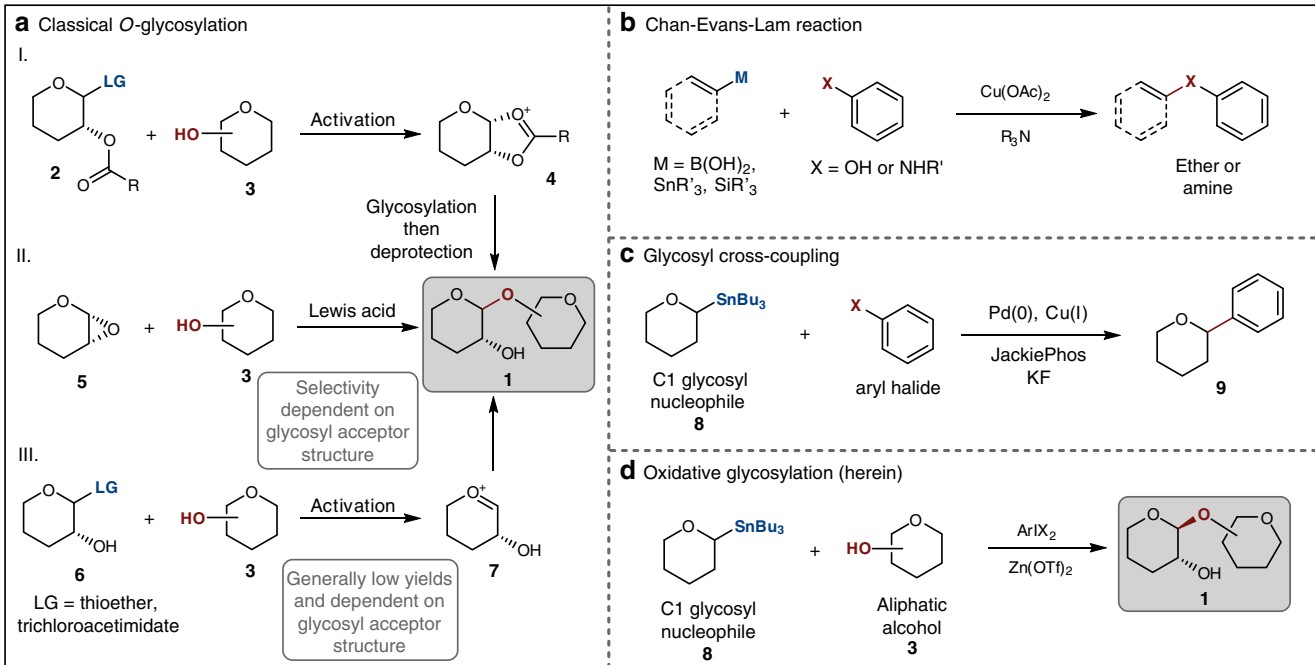

**Fig. 1** Strategies for O-glycosylation and the stereochemical outcome. **a** (i) Neighboring group participation (2-O-acyl glycosyl donor **2**) directed stereoselective glycosylation. Formation of dioxolenium cation **4** leads to 1,2-trans product using this method. LG leaving group (halogen, acetimidate, sulfide, etc.). (ii) O-glycosylation via a Lewis acid-catalyzed ring opening of a glycal epoxide. The epoxide is typically prepared from the oxidation of glycal and undergoes ring opening, resulting in a 2-hydroxy 1,2-trans glycoside. (iii) O-glycosylation with glycosyl donors bearing free hydroxyl group(s). The yield of this method is diminished due to the self-conjugation, and a mixture of anomers is usually obtained. LG leaving group (halogen, acetimidate, sulfide, etc.). **b** Etherification by the Chan−Evans−Lam reaction. Two nucleophiles are joined together to form an aryl ether. M main group metalloids (boronic acids, boronates, stannanes, silanes, and siloxanes). **c** A stereoselective C-glycosylation with a C1 nucleophile catalyzed by a transition metal catalyst. **d** A stereoselective O-glycosylation with glycosyl donors bearing free a hydroxyl group catalyzed by zinc triflate

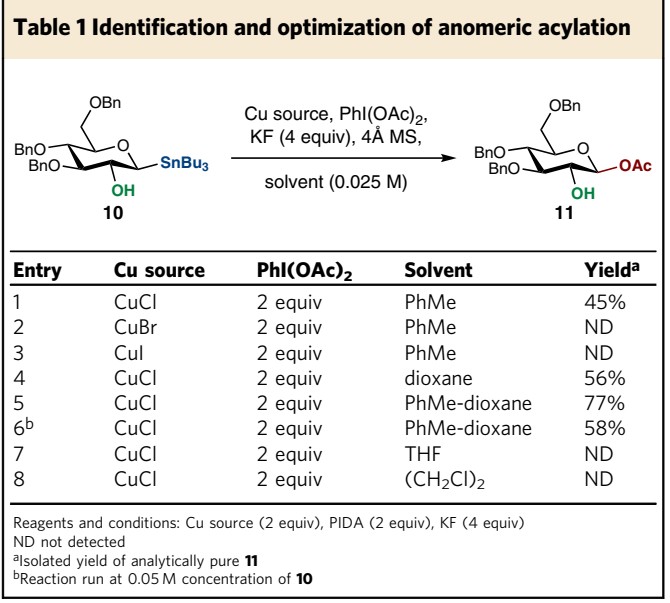

**Table 1 Identification and optimization of anomeric acylation**

| Entry | Cu source | PhI(OAc)$_2$ | Solvent | Yield[a] |
|-------|-----------|--------------|---------|----------|
| 1 | CuCl | 2 equiv | PhMe | 45% |
| 2 | CuBr | 2 equiv | PhMe | ND |
| 3 | CuI | 2 equiv | PhMe | ND |
| 4 | CuCl | 2 equiv | dioxane | 56% |
| 5 | CuCl | 2 equiv | PhMe-dioxane | 77% |
| 6[b] | CuCl | 2 equiv | PhMe-dioxane | 58% |
| 7 | CuCl | 2 equiv | THF | ND |
| 8 | CuCl | 2 equiv | (CH$_2$Cl)$_2$ | ND |

Reagents and conditions: Cu source (2 equiv), PIDA (2 equiv), KF (4 equiv)
ND not detected
[a] Isolated yield of analytically pure **11**
[b] Reaction run at 0.05 M concentration of **10**

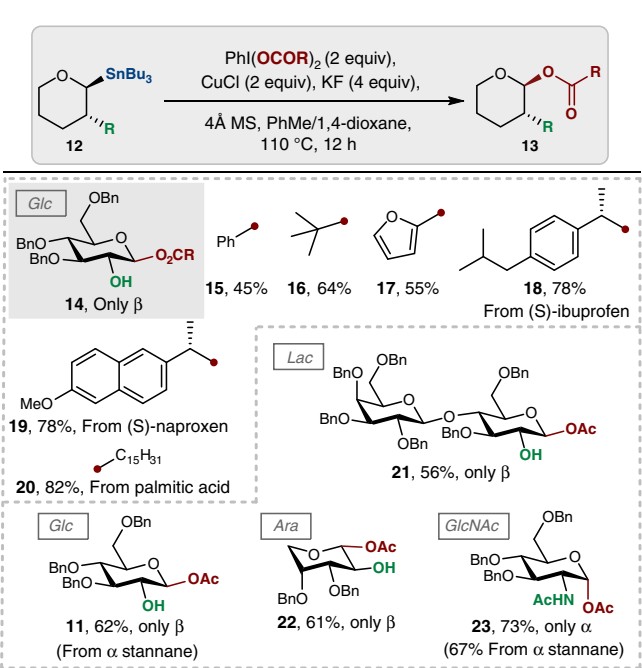

**Fig. 2** Scope of oxidative acylation of glycosyl C1 stannanes. In all cases, the formation of only one anomer was observed (based on the [1]H NMR analysis of the unpurified reaction mixtures). Phenyliodonium dicarboxylates were obtained from iodosobenzene diacetate and the corresponding carboxylic acid

engaging glycosyl donors with free hydroxyl group(s) (Fig. 1a)[4,5,22]. The use of glycosyl donors with conventional leaving groups such as in Schmidt donors and thioglycosides afforded a mixture of anomers although the hydroxyl group in the glycosyl donors often shows reduced reactivity that enables the site-selective control of the glycosylation event. Nevertheless, in glycosylation with donors such as anomeric dithiocarbamate and phosphate, it has been reported that free hydroxyl group can be tolerated without compromising the yield and stereoselectivity[23–25]. Somewhat related, although mechanistically distinct, is the photoredox-based approach with a boronic acid as an additive with little improvement in terms of stereoselectivity[26]. More recently, temporal boron-protection of thioglycoside donors with stoichiometric amount of dialkylboryl was shown to yield high 1,2-*trans* anomers[27]. Other strategies for the chemical *O*-glycosylation of unprotected glycosyl donors include chemoenzymatic synthesis, solid-phase approach, and catalyst-induced methods[28,29]. However, despite clear benefits of the approaches that rely on minimal protective group manipulations, previous attempts can seldom be applied to a broad substrate selection for regioselective installation of the glycosidic bond. Because of these inherent limitations, an approach that capitalizes on reactions with anomeric nucleophile can offer a promising solution. This proposal is based on the observation that under oxidative conditions nucleophiles such as alcohols, phenols, and amines can be coupled with boronic acids or stannanes forming a new C−O and C−N bond (the Chan−Evans−Lam reaction, Fig. 1d)[30–32]. However, the application of the oxidative coupling to chemical glycosylation is challenged by a limited knowledge about reactions of C($sp^3$) nucleophiles, as well as the unpredictability of stereochemical course in the C−O bond-forming step[33–36]. Phenols and amines with suitable nucleophilicity are optimal substrates for the coupling with C($sp^2$) partners and only isolated examples of cross-coupling of trifluoroborates with aliphatic alcohols have been reported[37,38].

We recently described a mechanistically unique method for the *C*-glycosylation of glycosyl stannane donors bearing free hydroxyl groups that features a remarkably high stereospecificity and minimal dependence on directing groups and carbohydrate scaffold[39,40]. C1 stannanes can be prepared for a range of mono- and oligosaccharides and retain their configurational stability after exposure to water, air, and even elevated temperatures.

Based on this result, here we show a conceptually distinct approach for stereoselective *O*-glycosylation as depicted in Fig. 1d. We hypothesized that under oxidative conditions anomeric stannanes might undergo transmetalation to copper or a ligand transfer to other oxidants such as hypervalent iodine[41,42] followed by reductive elimination to establish a new glycosidic bond. In this mechanistic paradigm, the C−O bond-forming step could proceed inter- or intramolecularly, depending on the nature of the oxidant and the nucleophile used. However, the ability to access this type of glycosyl donors opens the opportunity to perform these reactions with partially protected glycosyl donors.

## Results

**Optimization of reaction conditions**. To test our hypothesis, we first set out to identify conditions which could promote the union of alcohols with C1 nucleophiles without concomitant oxidation of the hydroxyl groups. Inspired by the prior work by Gin[43], we investigated reactions of anomeric stannanes in the presence of a stoichiometric oxidant (PhI(OAc)$_2$, PIDA), a copper catalyst, and KF as an additive facilitating the transmetalation step (Table 1). D-glucose nucleophile **10** was selected as a model system to gain a better understanding of the reactivity of anomeric stannanes under these conditions. We found that **10** could be converted into anomeric acetate **11** with exclusive 1,2-*trans* selectivity when treated with the oxidant (entry 1). Further optimizations revealed the unique role of CuCl and other Cu(I) salts were ineffective in promoting oxidation of **10** (entries 2 and 3). We established that 1,4-dioxane (entry 4) or a 1:1 mixture of PhMe and 1,4-dioxane (entries 5 and 6) resulted in the best isolated yield whereas THF or chlorinated solvents suppressed the acylation reaction (entries 7 and 8).

Because the iodine oxidant is the source of the acetate group, we next wondered if other carboxylate groups could be transferred into the anomeric position (Fig. 2). We found that

**Table 2 Optimization of oxidative glycosylation with anomeric nucleophiles**

| Entry | Substrate | Oxidant | Catalyst | Yield[a] | dr[b] |
|---|---|---|---|---|---|
| 1 | **10** | PhI(OAc)$_2$ | — | <1% | ND |
| 2 | **10** | PhIO | — | <1% | ND |
| 3[c] | **10** | PhIO | Tf$_2$NH | 35% | Only β |
| 4[c] | **10** | PhIO | TfOH | 65% | Only β |
| 5 | **10** | PhIO | Cu(OTf)$_2$ | 60% | Only β |
| 6[d] | **10** | PhIO | Zn(OTf)$_2$ | 86%[e] | Only β |
| 7 | **10** | PhIO | AgOTf | 31% | Only β |
| 8 | **10** | PhIO | Sc(OTf)$_3$ | 80% | Only β |
| 9 | **10** | PhIO | Zn(NTf$_2$)$_2$ | 31% | Only β |
| 10 | **24** | PhIO | Zn(OTf)$_2$ | <1% | ND |

ND not determined
[a]NMR yield using internal standard (CHBr$_3$)
[b]Based on the $^1$H NMR analysis of the unpurified reaction mixture
[c]10 mol% of acid catalyst
[d]**25** was obtained in 57% (only β) from the corresponding α-stannane using conditions from entry 6
[e]83% isolated yield

aromatic (**15**), heteroaromatic (**17**), aliphatic (**16**, **20**), and complex (**18**, **19**) ester groups could be installed at the C1 position using the corresponding carboxylic acids in high selectivities. To further probe the generality of this reaction, we applied the optimized conditions from Table 1 to reactions forming anomeric acetates of D-lactose (**21**), D-arabinose (**22**), and D-glucosamine (**23**). The configuration of the anomeric nucleophile has no bearing on the stereochemical outcome of acylation reactions, as demonstrated in the preparation of **11** and **23**, using the α- and β-anomers, respectively, although the yields of these reactions are slightly diminished.

Having established a general set of conditions for the synthesis of anomeric esters, we next investigated reactions promoting the formation of the glycosidic bond between anomeric nucleophiles and alcohols (Table 2). When **10** was exposed to the optimized conditions from Table 1 using alcoholic solvents (MeOH, *i*-PrOH, or CyOH), acetate **11** was formed as the only product. In the absence of CuCl, no reaction was observed with **10** using PIDA or PhIO (Table 2, entries 1 and 2). Based on these results, we hypothesized that the transfer of the iodine into the anomeric position could be promoted by a Lewis or Brønsted acid. To test this proposal, β-D-glucose **10** was reacted with PhIO in the presence of catalytic amounts of Tf$_2$NH (entry 3) or TfOH (entry 4) and the reaction resulted in a clean conversion into glycoside **25** within 12 h at room temperature.

Further studies revealed the critical role of the catalyst and the oxidant in promoting *O*-glycosylation. We were pleased to find that the combination of PhIO and Cu(OTf)$_2$ or Zn(OTf)$_2$ (but not ZnCl$_2$ and ZnBr$_2$, entries 5 and 6) promote oxidative reaction with anomeric nucleophiles at room temperature with high β selectivity ($^1$H NMR). Other stronger oxidants (PhI(OTf)$_2$, NFSI, XeF$_2$, Br$_2$, CAN, Oxone®, *m*-CPBA) resulted only in decomposition of the anomeric stannane under a variety of conditions, including those buffered by a bulky base (2,6-di-*t*-butylpyridine). The presence of a metal triflate is required for the reaction to take place and alkaline metal salts (NaOTf, KOTf) resulted in no reaction (recovered **10**) even under forcing conditions (100 °C,

24 h). Catalyst loadings of Zn(OTf)$_2$ as low as 1% could promote the conversion of **10** into **25**, although after 2 days the anomeric nucleophile was not fully consumed indicating that the rate of the reaction could be increased by simply adding more catalyst. For practical reasons, however, 5 mol% of Zn(OTf)$_2$ was used as the standard conditions for the subsequent studies. Silver and scandium triflates (entries 7 and 8) as well as Zn(NTf$_2$)$_2$ (entry 9) are viable catalysts although the yields with these promoters were suboptimal. There is also a negligible effect of temperature on the reaction yield although elevated temperatures (>100 °C) have detrimental effects. Finally, the optimal solvents for this transformation are CHCl$_3$ or aromatic solvents such as PhMe and benzene. Solvents that are known to participate in the stabilization of the oxonium intermediate (1,4-dioxane, THF, Et$_2$O, MeCN) result only in low yields (<10%). For all conditions tested with **10**, exclusive formation of the 1,2-*trans* anomer **25** was observed even when the α-stannane was used as the substrate, which is in striking contrast to the stereospecific C−C bond-forming reactions under Pd-catalyzed conditions. Curiously, the presence of a free hydroxyl group at C2 is necessary for the reaction to take place and exposure of the substrate **24** with protected C2-OH position resulted only in the recovery of the stating material (entry 10).

**Substrate scope and synthetic applications**. The optimized conditions from Table 2 were then applied to reactions with various alcohol acceptors and glycosyl nucleophiles (Fig. 3). We found that the D-glucose stannane **10** gave consistently high yields and β selectivities when reacted with small alcohol nucleophiles (**27**−**31**) without the formation of aldehyde or ketone by-products resulting from oxidation of the alcohols by iodine(III) reagents. Similarly, high yields were recorded for reactions with D-glucose acceptors (**32**−**40**), D-galactose (**41** and **42**), deoxy sugars (**42**-**46**), and reactions with disaccharide stannanes (**47** and **48**). Common protecting groups such as 2-naphthylmethyl (**34**), *para*-methoxybenzyl (**35**), and levulinyl (**37**) were tolerated and resulted in moderate to excellent yields, providing more options for orthogonal protecting group manipulations for synthesizing complex carbohydrates. To our delight, diacetone-D-glucose, containing the acid-sensitive acetal group, can also be coupled in 86% yield under the oxidative reaction conditions forming disaccharide **39**. In general, glycosyl acceptors equipped with electron-withdrawing groups (**36**−**38**) are also viable substrates for the oxidative glycosylation without erosion of selectivity. When the reaction was attempted with congested acceptors, we found that a more reactive oxidant (Koser's reagent, PhI(OH)OTs[44] or 3,5-(CF$_3$)$_2$C$_6$H$_3$I(OH)OTs)[45] gave higher yields than PhIO and effectively formed glycosides with C4-OH and C2-OH acceptors (**49** and **50**). We attribute the improved yields with hydroxyiodoaryl reagents to the ease of activation of the iodine and monomeric structures of these oxidants as opposed to the polymeric nature of iodosobenzene. Unlike reactions of partially protected anomeric thioethers[5], trichloroacetimidates[22], and glycal epoxides[17] with hindered glycosyl acceptors that result in a mixture of anomers, oxidative conditions promote exclusive 1,2-*trans* selectivity without the need for transient protection of the C2 position.

The oxidative glycosylation is easy to conduct: all substrates are indefinitely stable, the reactions are performed at room temperature, and, due to the significant difference in polarity between the tin by-products and glycosides, purification by column chromatography on silica gel is straightforward and efficient. Furthermore, various technologies are available to remove trace amounts of tin and transition metals[46,47].

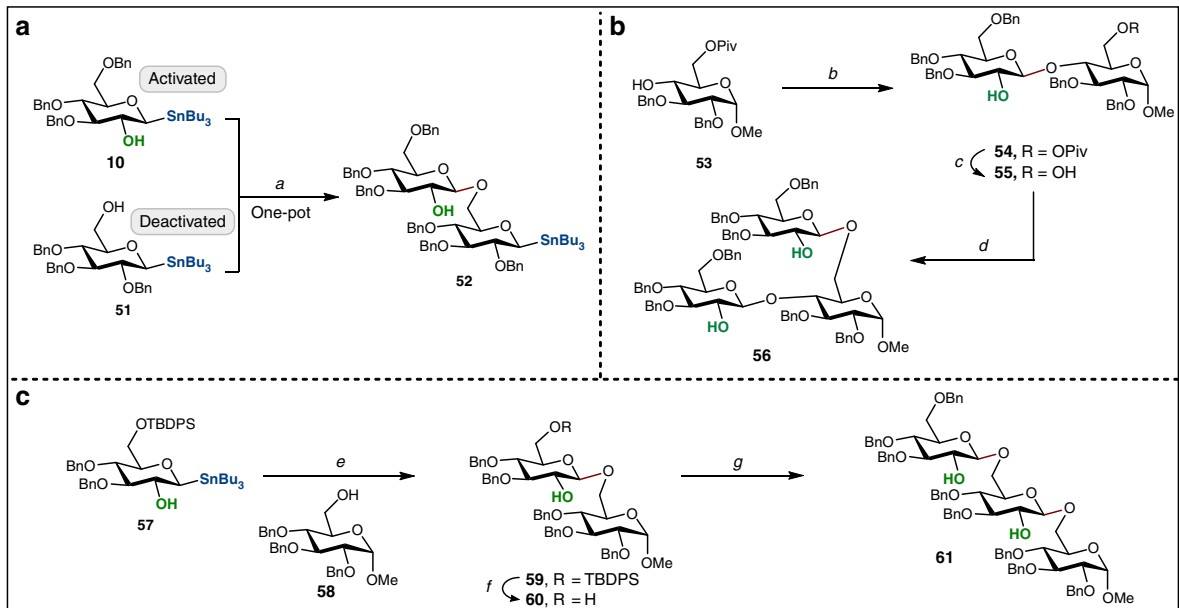

**Fig. 3** Scope of oxidative glycosylation with anomeric stannanes. In all cases, only the formation of 1,2-*trans* anomer was observed (based on the [1]H NMR analysis of the unpurified reaction mixtures). Conditions **a**: compounds **27–31**, **41**, and **44** and **47**. Conditions **b**: compounds **32–39**, **40**, **42**, **43**, **45**, **46**, **48** using PhI(OH)OTs and compounds **49** and **50** using 3,5-(CF$_3$)$_2$C$_6$H$_3$I(OH)OTs. Compound **39** required the addition of 2,4,6-tri-*tert*-butylpyrimidine

**Fig. 4** Orthogonal and sequential glycosylations. **a** PhI(OH)OTs, Zn(OTf)$_2$, CHCl$_3$, 23 °C, 36 h, 43%; **b 10**, PhI(OH)OTs, Zn(OTf)$_2$, CHCl$_3$, 23 °C, 36 h; **c** MeONa, MeOH, 50 °C, 10 h, 55% over two steps; **d 10**, PhI(OH)OTs, Zn(OTf)$_2$, CHCl$_3$, 23 °C, 48 h, 56%; **e** PhI(OH)OTs, Zn(OTf)$_2$, CHCl$_3$, 23 °C, 36 h, 82%; **f** TBAF, THF, 23 °C, 24 h, 88%; **g 10**, PhI(OH)OTs, Zn(OTf)$_2$, CHCl$_3$, 23 °C, 36 h, 71%

From the optimization studies, we determined that the a C2-coordinating group is necessary for successful *O*-glycosylation. From this observation, we hypothesized that the oxidative protocol could be applied in orthogonal glycosylation with two anomeric nucleophiles in which the free hydroxyl group at C2 activates one nucleophile toward oxidation while the C2 position in the second anomeric nucleophile is protected thus allowing it to serve as the glycosyl acceptor. To this end, stannanes **51** and **10**

**Table 3 Oxidative transformations of D-glucose 10 and epoxide 62**

| Entry | Substrate | PhI(OH)OTs | Zn(OTf)$_2$ | 63[a] | 64[a] |
|---|---|---|---|---|---|
| 1 | 10 | Yes | Yes | 41% | 43% |
| 2 | 10 | Yes | No | 36% | 43% |
| 3 | 63 | Yes | Yes | 18% | ND |
| 4 | 63 | Yes | No | 25% | ND |
| 5[b] | 63 | Yes | Yes | ND | ND |
| 6[c] | 63 | Yes | No | ND | ND |

10 or 62 (1 equiv), PhI(OH)OTs (1 equiv), Zn(OTf)$_2$ (5 mol%), CHCl$_3$ (0.05 M), 23 °C, 3–36 h
ND not determined
[a]NMR yield using internal standard (CHBr$_3$)
[b]As entry 3 with 2 equiv of CyOH; 25 was formed in 51%
[c]As entry 4 with 2 equiv of CyOH; 25 was formed in 53%

were merged under the standard conditions to form saccharide 52 in 43% with excellent β stereoselectivity with the remainder of the isolated material being D-glucal (Fig. 4a). The sequence of reactions in Fig. 4a represents an example of an orthogonal and sequential union of anomeric nucleophiles based on the activating group at C2. Alternatively, the distinct reactivities of primary and secondary alcohols implies the possibility of chemoselective glycosylation which we demonstrated in the context of sequential glycosylations (Fig. 4b). Thus, the C6-protected glucosyl stannanes 53 and 10 were joined together under the standard protocol to form disaccharide 54, followed by deprotection of the pivaloyl ester group. The installation of the third monosaccharide was accomplished in a stereoselective fashion and resulted in the preparation of trisaccharide 56 in 56% yield. Furthermore, linear trisaccharide 61 was also obtained with exclusive 1,2-trans selectivity further demonstrating our method's utility in the syntheses of longer oligosaccharides involving minimal protection group operations (Fig. 4c). In contrast to other paradigms in orthogonal glycosylation that capitalize on a combination of thioglycosides and anomeric fluorides or bromides[48,49], our method features excellent selectivity and milder reaction conditions (e.g., ambient reaction temperatures) which allows for the potential translation of this methodology to the automated assembly of complex oligosaccharides.

**Mechanistic investigations**. The significant difference in stereochemical outcome between the oxidative glycosylation and that of other transformations affording 1,2-trans pyranosides led us to carry out a series of control experiments summarized in Table 3 and Fig. 5. The following results were obtained: in the absence of an alcohol nucleophile, stannane 10 was converted into almost equimolar amounts of formate 63 and D-glucal 64 (entries 1 and 2). This reaction is independent from the presence of a Lewis acid additive (for details, see Supplementary Figure 1). To investigate the intermediacy of a glycal epoxide in oxidative glycosylation, 62 was treated with Koser's reagent and resulted only in oxidative cleavage product 63 (entries 3 and 4). Since the activating effect of tributyltin ether in O-glycosylation was reported by

Danishefsky[21], 62 was treated with MeOSnBu$_3$ (2 equiv) and Zn (OTf)$_2$ (5 mol %) in CH$_2$Cl$_2$. Although the same stereoselectivity (α:β < 1:99) and a slightly diminished yield (79%) were observed, the epoxide opening reaction with stannyl ether was significantly slower (>6 h) than the glycosylation with stannane 10 (1 equiv), methanol (2 equiv), Koser's reagent (1.2 equiv), and 5 mol% of Zn (OTf)$_2$ (<20 min, 86%, compound 27). Schmidt reported glycosylation of 2-hydroxy-1-acetimidates resulting in a mixture of anomers[22]. Similar results were reported by Baker in reactions of 2-hydroxy-1-thioglycosides.[50] To probe the existence of a glycal epoxide, the two reactions described above were monitored with $^1$H NMR. While H1 and H2 in 62 showed two well-defined peaks and decayed over time, the same signals were absent in the reaction with stannane 10, Koser's reagent, and Zn(OTf)$_2$. In the presence of CyOH (2 equiv) and Koser's reagent, O-glycoside 25 was formed in modest yield with no detectable amounts of 63 and 64 (entries 5 and 6). A mechanism involving the oxidative functionalization of glycals via iodoacetoxylation[43] was excluded on the basis of the reaction of D-glucal 64 with Koser's reagent and CyOH resulting in only 8% of 25 and unidentified compounds. Both α- and β-stannanes of D-glucose afforded only the 1,2-trans O-glycoside 25 regardless of the configuration of the substrate. A reaction of 3,4,6-tri-O-benzyl-D-glucose (a 1,2-diol) with PhIO or Koser's reagent resulted in oxidative cleavage of the diol and exclusive formation of 63 indicating that a cyclic λ$^3$-iodane is not a viable intermediate in the C—O bond-forming step[16]. A coordinating group at C2 is necessary for the reaction to take place indicating that the delivery of the oxidant (e.g., ArIO or ArIX$_2$) into the anomeric position is initiated by a ligand exchange. A reaction of 10 with Koser's reagent monitored by MS-ESI showed a molecular ion at m/z 945.2 corresponding to protonated iodonium species 65 (Fig. 5a). In the same reaction mixture, tin tosyalte 66 was also detected (vide infra). Furthermore, exposure of fully protected tetra-O-benzyl-β-D-glucose stannane to PhIO or Koser's reagent resulted in recovery of the starting material after 24 h at room temperature. Similar observations were recorded for 2-deoxy-tri-O-benzyl-β-D-glucose stannane suggesting that the steric hindrance at C2 plays a negligible role in initiating the transfer of the iodine group. Lewis or Brønsted acids are necessary for the reaction to take place supporting our initial hypothesis that the hypervalent iodine reagent undergoes activation toward nucleophilic addition and/or substitution.

Further studies were focused on reaction analysis using nuclear magnetic resonance (NMR) spectroscopy to corroborate the identity of reactive intermediates. Treatment of anomeric stannane 10 (1 equiv) with Koser's reagent (1.2 equiv) in the presence of a Lewis acid (Zn(OTf)$_2$, 1 equiv) was analyzed by low-temperature NMR (−80 to 20 °C). The $^1$H NMR spectra for the reaction mixture revealed a clear doublet at 5.91 ppm and $^3J_{(HH)}$ of 3.5 Hz corresponding to the axial substitution (Fig. 5b). Identical results were observed when α-stannane and Koser's reagents were reacted under the same conditions. When the temperature was raised slowly, the characteristic proton signal gradually shifted downfield by 0.05 ppm and disappeared around 0 °C after 15 min. The use of an analog of Koser's reagent where the iodophenyl group was modified with an electron-withdrawing group (m-CF$_3$) resulted in no change in $^1$H NMR spectra and revealed the formation of the same intermediates as with Koser's reagent. However, exchange of the tosyl group into benzenesulfonate or 2,4-dinitrobenzenesulfonate resulted in the formation of distinct intermediates with an α-configuration at 5.96 ppm ($^3J_{(HH)}$ = 3.6 Hz) and 6.32 ppm ($^3J_{(HH)}$ = 3.4 Hz), respectively. In an independent experiment, we also generated the anomeric triflate of 3,4,6-tri-O-benzyl-D-glucose that showed a distinct peak at 6.38 ppm ($^3J_{(HH)}$ = 3.5 Hz) which was not observed in the

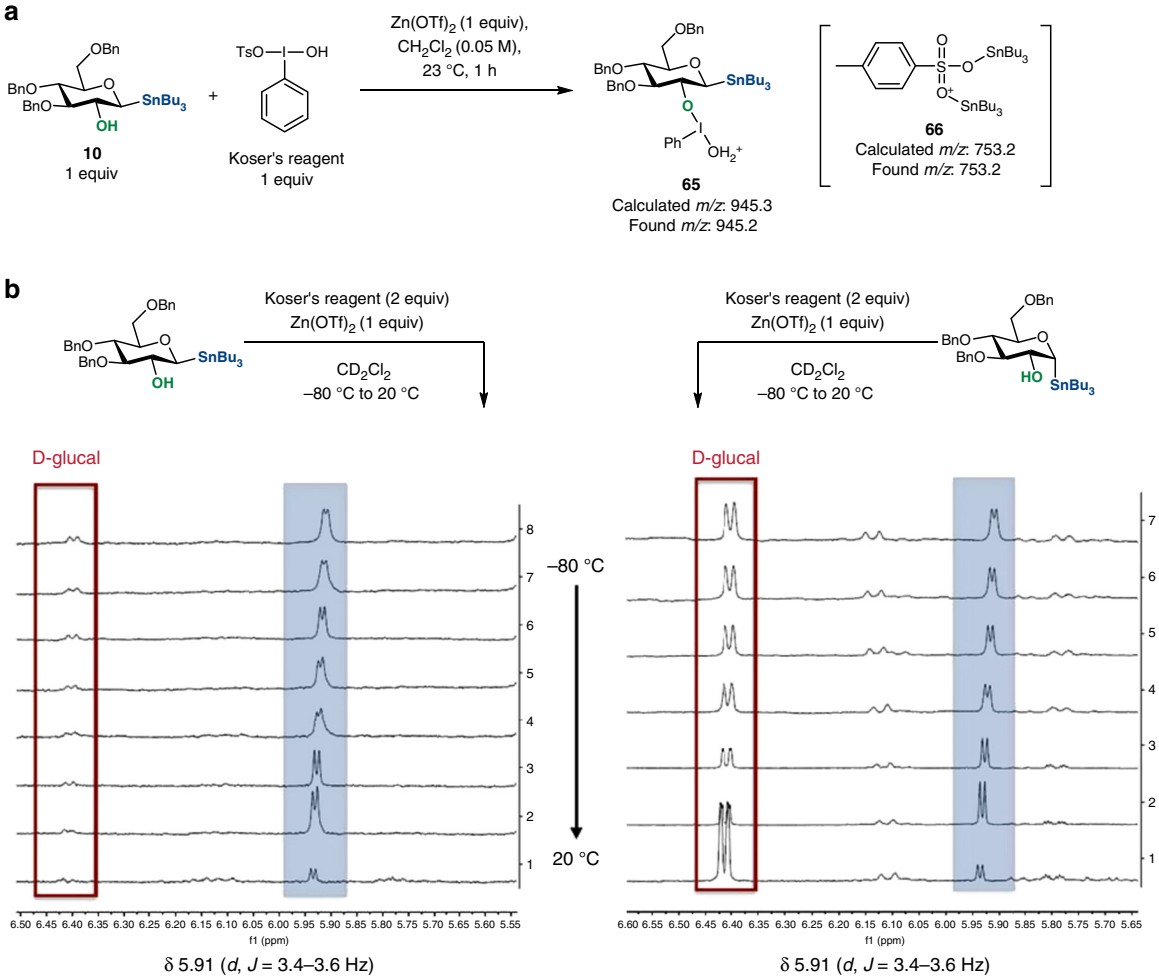

**Fig. 5** Mechanistic investigations. **a** Reaction of stannane **10** with Koser's reagent. **b** $^1$H NMR spectra of reaction mixtures of α- and β-stannanes and Koser's reagent

reactions with Koser's reagent. The similarities of the observed values in reactions with hypervalent iodine to the previously reported data of anomeric α-tosylate suggest that the stereochemical determinant in this reaction manifold is an α-sulfonate[51,52]. Anomeric tosylates have been shown to undergo $S_N2$ reactions with oxygen nucleophiles[51]. These findings are in agreement with the control experiments mentioned above indicating that the α-sulfonate is a more reactive species than the glycal epoxide and provides the 1,2-*trans* stereochemistry in the products.

Taken together, the mechanistic studies suggest that the C2-hydroxyl group initiates the oxidative glycosylation and modulates the delivery of the hypervalent iodine oxidant. Although the intermediacy of an epoxide cannot be completely excluded at this point, the low yield of O-glycoside **25** observed in a reaction with **62**, exclusive anomeric selectivities for reactions with congested acceptors and with glycosyl donors known to result in a mixture of anomers when reacted in the form of the corresponding epoxides (e.g., galactopyranose), point to an alternative mechanism (Fig. 6). Based on the collected data, we propose that the reaction of **67** with ArIO or ArI(OH)OTs is initiated by a ligand exchange between the C2-OH and an iodine reagent[53]. The formation of intermediate **68** is catalyzed by a Lewis or Brønsted acid that activates ArIO toward addition of an alcohol or facilitates the displacement step at I(III) in the case of Koser's reagent. Given that PhIO exists as an oligomeric structure, a Lewis acid can also promote depolymerization of

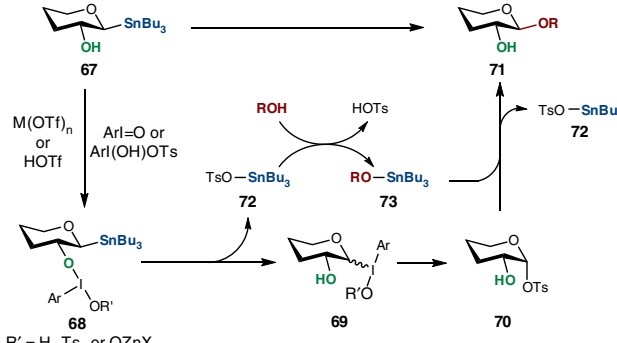

**Fig. 6** Mechanistic proposal of oxidative O-glycosylation. Plausible reaction mechanism based on experimental investigations

ArIO and catalyze dissolution of the polymeric oxidant. Next, a transfer of iodine into the anomeric position results in the transient formation of intermediate **69** which, driven by the propensity of iodine to return to its normal valency, undergoes substitution or reductive elimination to generate the α-sulfonate **70** (observed experimentally) followed by displacement to yield **71**. Solvolysis studies with a vinyl iodonium reagent show that the iodonium moiety is an excellent leaving group with a nucleofugal ability $10^6$ higher than the triflate ion (the "IAr(OR′)" group functions as a hypernucleofuge)[54,55].

Although our initial mechanistic hypothesis for the use of organotin compounds was based on the notion that anomeric stannanes are configurationally stable C1 nucleophiles, the presence of **66** suggests that the tin group, following its release during the transfer of iodine into the anomeric position, could function to activate the hydroxyl group in the glycosyl acceptor forming nucleophilic ether **73** that subsequently reacts with **70**. Alternatively, the hydroxyl group in glycosyl donor **67** could be activated with tin thus promoting a ligand exchange at the iodine atom of the oxidant. Taken together, the combination of tributyltin-activated glycosyl acceptor and the formation of an α-tosylate lead to high anomeric selectivities of the glycosylation reactions surveyed under the oxidative conditions.

## Discussion

Here, we show an oxidative O-glycosylation reaction involving anomeric nucleophiles with alcohols and carboxylic acids resulting in exclusive anomeric selectivities. These reactions require a C2-coordinating group and open possibilities for the development of preparative oligosaccharide protocols capitalizing on orthogonal activation modes of anomeric nucleophiles with minimal protective group manipulations. Furthermore, this method can serve as a complementary tool to classical O-glyco-sylation protocols and, given its mild conditions and ambient reaction temperatures, can be easily adapted to suit the needs of automated synthetic strategies.

## Methods

**Materials.** For $^1$H, $^{13}$C NMR spectra of compounds in this manuscript, see Supplementary Information. For detailed synthetic procedures, see Supplementary Information.

**General procedures.** General procedure for anomeric acylation: Under N$_2$, a one-dram vial was charged with anomeric stannane (0.100 mmol), phenyliodonium biscarboxylate (0.200 mmol), CuCl (0.200 mmol), KF (0.400 mmol), freshly activated powdered 4 Å MS (ca. 100 mg), anhydrous toluene (1.00 mL) and anhydrous 1,4-dioxane (1.00 mL) and was heated up to 110 °C. The reaction was stirred for 12 h, and filtered through Celite®. The filtrate was concentrated and purified by column chromatography on SiO$_2$.

General procedure for glycosylation with iodosobenzene: Under N$_2$, anomeric stannane (0.100 mmol−0.200 mmol), the corresponding alcohol (0.100−0.200 mmol), Zn(OTf)$_2$ (0.010−0.100 mmol), iodosobenzene (0.200−0.300 mmol), freshly activated 4 Å MS and anhydrous CHCl$_3$ (0.50−2.00 mL) were successively added into a vial. The reaction mixture was stirred at room temperature for the indicated period of time, filtered through a pad of silica gel, and concentrated. $^1$H NMR spectra were recorded using this mixture to evaluate diastereoselectivity. The crude material was purified by column chromatography on SiO$_2$.

General procedure for glycosylation with hydroxy(tosyloxy)iodobenzene (Koser's reagent): Under N$_2$, anomeric stannane (0.100 mmol), the corresponding alcohol (0.100 mmol), Zn(OTf)$_2$ (0.005 mmol), hydroxy(tosyloxy)iodobenzene (0.100 mmol), freshly activated 4 Å MS, and anhydrous CH$_2$Cl$_2$ (4.00 mL) were successively added into a vial. After stirring at room temperature for 12 h, anomeric stannane (0.100 mmol) and hydroxy(tosyloxy)iodobenzene (0.100 mmol) were added and stirred for additional 12 h, and this procedure was repeated one more time. The reaction mixture was filtered through a pad of silica gel, and concentrated. $^1$H NMR spectra were recorded using this mixture to evaluate diastereoselectivity. The crude material was purified by column chromatography on SiO$_2$.

## Data availability

All relevant data are available upon request from the authors.

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

## Acknowledgements

This work was supported by the University of Colorado at Boulder and the NIH (GM125284). Mass spectral analyses were recorded at the University of Colorado Boulder Central Analytical Laboratory Mass Spectrometry Core Facility (partially funded by the NIH, RR026641).

## Author contributions

T.Y. developed the method, performed the experiments on the oxidative acylation and glycosylation of anomeric stannanes, and conducted the spectroscopic studies. F.Z. performed the experiments on the oxidative glycosylation of anomeric stannanes and mechanistic studies. M.A.W. conceived the study and supervised the experimental work. All authors participated in the design of the study and interpretation of the results. M.A.W. wrote the manuscript with contributions from all authors. All authors have read and approved the final version of the manuscript.

## Additional information

**Competing interests:** The authors declare no competing interests.

