## [Peer Review File · Nature Communications]

Reviewer #1 (Remarks to the Author):

The manuscript describes a method for formation of O-glycosidic bonds by oxidative coupling of anomeric trialkylstannanes with alcohols. The transformation employs hypervalent iodine reagents (PhIO or ArI(OH)OTs) as oxidants along with Zn(II) trifluoromethanesulfonate as a Lewis acid catalyst. A free OH group at the 2-position of the anomeric stannane donor is needed for efficient glycosylation reactivity. A copper-mediated synthesis of beta-configured anomeric esters from anomeric stannanes and (dicarboxyiodo)benzene is also demonstrated. The coupling with alcohols delivers beta-gluco- and beta-galacto-configured products with a free 2-OH group, and is tolerant of a range of acceptors, including sterically hindered alcohols and protected carbohydrates. The requirement for a free 2-OH group has been used to achieve selective activation of one anomeric stannane over another.

This is interesting new reactivity of anomeric stannanes that builds on the corresponding author's previous research on C-glycosylations employing such reaction partners (refs. 34, 35). However, I have concerns regarding the way that the authors have 'framed' this work in relation to previous research on glycosylations involving glycal epoxides. A major revision of the manuscript would be needed prior to further consideration for publication in Nature Communications.

As the authors have pointed out, the primary way to generate 1,2-trans-configured gluco- or galactopyranosides having a free 2-OH group is by ring-opening of glycal epoxides. Although sterically hindered alcohols do present challenges in certain instances, several examples employing such acceptors have been reported (e.g., Di Bussolo, Kim and Gin, *J Am. Chem. Soc.* 1998, 120, 13515-13516; Honda and Gin, *J. Am. Chem. Soc.* 2002, 124, 7343-7352; I would argue that these papers from the Gin group are more relevant to the present work than the one cited at ref. 19, which describes alpha-mannopyranoside synthesis). On the whole, I am concerned that the introduction overstates some of the limitations of the glycal epoxide approach, and a more balanced description (along with appropriate literature citations) would be preferable.

My other, and more important, concern, relates to whether this is truly a 'conceptually orthogonal approach' (page 3) relative to the methods that employ glycal epoxides. I agree that the experiments described on page 12 support a mechanism in which the free 2-OH group serves to accelerate (in the presence of a Lewis or Bronsted acid) the conversion of the anomeric stannane into a leaving group by the action of the hypervalent iodine reagent. However, this still leaves open the possibility that glycal epoxides are generated as intermediates after activation of the stannane. It is not clear why intermolecular attack by a glycosyl acceptor should compete with cyclization by the 2-OH group upon formation of an intermediate such as 59. The experiments summarized in Table 3 do not rule out the formation of a glycal epoxide from the anomeric stannane (and indeed, entries 5 and 6 confirm that such a glycal epoxide undergoes relatively efficient glycosylation under these reaction conditions). The fact that the stannane-based couplings show such high 1,2-trans stereoselectivities

is used as indirect support for a pathway that does not involve glycal epoxides, but this is a dangerous line of logic because data for the corresponding couplings of glycal epoxides under identical conditions do not exist. Not only are the Lewis acid catalysts not the same (e.g., gold(I) triflate in ref. 20, scandium(III) triflate in an alpha-selective example from the Gin 1998 article cited above), but the present system includes a triorganotin byproduct that may play a role by generating a stannyl ether, as suggested by the authors on page 14. In fact, couplings of stannyl ethers with glycal epoxides have been reported (e.g., Randolph and Danishefsky, *J. Am. Chem. Soc.* 1995, 117, 5693-5700), and provide a significant improvement over the reactions with the corresponding alcohols when hindered acceptors are used. This work should also have been cited. A fair comparison of the present work with glycal epoxide couplings would thus require the use of acceptor-derived stannyl ethers as nucleophiles in the presence of $\text{Zn}(\text{OTf})_2$. Overall, the data provided to this point do not allow for firm conclusions to be made as to whether this is truly a unique approach towards glycosylation or simply a new way to generate glycal epoxides, potentially with concurrent glycosyl acceptor activation via stannyl ether formation.

This is a pertinent issue because the stannane donors are synthesized from the glycal epoxides, and so it would be important to know whether a direct coupling of the glycal epoxides with stannylated acceptors would suffice in some instances. But it is not to say that the chemistry described here is not a useful and interesting way of generating glycosidic linkages. From my perspective, it would be preferable to focus on the potential advantages of this method in preparative carbohydrate chemistry, and to highlight the new aspects of reactivity (namely, the OH-accelerated substitution of the anomeric centre) without attempting to describe this work as being fundamentally different from glycal epoxide activation. If the authors wish to take a position on this more fundamental difference, then more experimental work aimed at ruling out glycal epoxide intermediates would be needed.

Minor points:

1. What evidence do the authors have to support the assignment of the major conformer of products 39 and 40? The destabilizing 1,3-diaxial interaction between oxygenated substituents should more than overcome any favorable anomeric effect that might favor the $4C_1$ conformation for this type of alpha-arabinopyranoside. Assignments of signals in the ^1H NMR spectra of these compounds were not provided, so it was not clear that the major conformer was assigned based on coupling constant analysis. Incidentally, the structures of these compounds in the SI (1,2-cis) are not consistent with those shown in the paper (1,2-trans).
2. It would be preferable to restrict the Figure captions to a simple description of what is depicted, rather than including interpretations of the results (e.g., 'great yields and remarkably high stereospecificities', Figure 1 caption) or generalizations that merit more detailed discussion (e.g., 'When .. ROH is more hindered than a primary alcohol, a mixture of anomers is obtained', Figure 1 caption).

Reviewer #2 (Remarks to the Author):

Recommendation: The manuscript does not meet the requirements of urgency and novelty that justify publication in Nature Commun.

In the submitted manuscript Walczak and colleagues report conditions for doing efficient stereoselective oxidative glycosylation of anomeric nucleophiles with alcohols and carboxylic acids. After careful evaluation of manuscript, I do not find the manuscript suitable for publication in Nature Commun, due to the following reasons.

1. There are relatively many reports of this type transition metal catalyzed O-glycosylation (specially with the metal triflates) in the literature but the logic of exploring these tin-derivatives is weak. So I haven't found enough significance from the chemistry outlook.

2. The reaction between tin-glycoside with alcohols and carboxylic acids is bit predictable without doubt. In fact, The authors have also done similar kind of work before (J. Am. Chem. Soc., 2016, 138 (37), 12049–12052; J. Am. Chem. Soc., 2017, 139, 17908–17922). The authors haven't justified enough the novelty of their work particularly the difference from the previous reports.

3. What authors claimed in the text is the involvement of 2-hydroxyl group with hypervalent iodine, in tin-glycoside during glycosylations. It is eye-catching, but the mechanistic study for the reaction is not sufficient enough that could demonstrate the mechanistic pathway.

Thus, I find this too weak for publication in such a high impact journal like Nature Commun. In my opinion, manuscript would be more suitable for a more specialized journal or a general chemistry journal with slightly less stringent requirements than Nature and I believe it will be more suitable in journals like ACS Catalysis or Chemical Science.

Reviewer #3 (Remarks to the Author):

Professor Walczak and coworkers have shown here an exciting work on chemical glycosylation by using glycosyl stannane donor and its new role in oxidative O-glycosylation. The manuscript is well-written and easy to understand. The supporting information such as NMR spectra are high quality, well-prepared and presented. My comments on this manuscript are;

1. The authors state that “A conceptually orthogonal approach to addresses the problem of poor anomeric stereoselectivities relies on the use of glycosyl donors with free hydroxyl groups that can act as directing groups, which will eliminate the need for the protection/deprotection steps. Herein, we report what is to the best of our knowledge, the first one-pot O-glycosylation starting from partially protected glycosyl donors under mild reaction conditions without compromising the yields and with complete anomeric selectivities”

However, there are previous reports on the stereoselective glycosylation using partially protected glycosyl donors such as glycosyl dithiocarbamates by Wei and coworkers or glycosyl phosphates by Seeberger and coworkers. Please check these references;

1. Wei et.al. *Org. Lett.* 14, 13, 3380-3383
2. Wei et.al. *J. Org. Chem.* 79, 6, 2611-2624
3. Seeberger et.al. *J. Am. Chem. Soc.*, 123, 9545–9554

2. A versatile glycosylation should allow chain propagation from both non-reducing and reducing end of carbohydrate. However, the authors have shown only one example of trisaccharide synthesis which is propagated from the non-reducing end.

3. The scope of O-glycosylation is limited to glycosyl donors and acceptors equipped with benzyl ether or ester group. Thus, I wonder about its application toward those less stable protecting groups which are necessary for the orthogonal protecting group manipulations for complex carbohydrates.

4. Authors should provide a synthesis of more complex oligosaccharide to emphasize practicality of this oxidative glycosylation.

5. For moderate yields on the glycosylation in Scheme 3, could authors give comment on the byproduct?

6. The use of glycosyl stannane itself may be challenging to others especially those who are not in a field of organometallic chemistry. Could authors give comment on the removal of organotin after the glycosylation.

7. Please check the consistency of reference format.

In summary, I will accept this manuscript if the authors could provide more examples to address on my comment number 3, and 4.

We would like to thank all reviewers for their time and valuable comments on our manuscript. We have performed additional experiments to address all concerns and our response is highlighted in blue. New experimental data relevant to the issues raised by the reviewers are included in the updated manuscript.

Reviewer #1 (Remarks to the Author):

The manuscript describes a method for formation of O-glycosidic bonds by oxidative coupling of anomeric trialkylstannanes with alcohols. The transformation employs hypervalent iodine reagents (PhIO or ArI(OH)OTs) as oxidants along with Zn(II) trifluoromethanesulfonate as a Lewis acid catalyst. A free OH group at the 2-position of the anomeric stannane donor is needed for efficient glycosylation reactivity. A copper-mediated synthesis of beta-configured anomeric esters from anomeric stannanes and (dicarboxyiodo)benzene is also demonstrated. The coupling with alcohols delivers beta-gluco- and beta-galacto-configured products with a free 2-OH group, and is tolerant of a range of acceptors, including sterically hindered alcohols and protected carbohydrates. The requirement for a free 2-OH group has been used to achieve selective activation of one anomeric stannane over another.

This is interesting new reactivity of anomeric stannanes that builds on the corresponding author's previous research on C-glycosylations employing such reaction partners (refs. 34, 35). However, I have concerns regarding the way that the authors have 'framed' this work in relation to previous research on glycosylations involving glycal epoxides. A major revision of the manuscript would be needed prior to further consideration for publication in NatureCommunications.

As the authors have pointed out, the primary way to generate 1,2-trans-configured gluco- or galactopyranosides having a free 2-OH group is by ring-opening of glycal epoxides. Although sterically hindered alcohols do present challenges in certain instances, several examples employing such acceptors have been reported (e.g., Di Bussolo, Kim and Gin, J Am. Chem. Soc. 1998, 120, 13515-13516; Honda and Gin, J. Am. Chem. Soc. 2002, 124, 7343-7352: I would argue that these papers from the Gin group are more relevant to the present work than the one cited at ref. 19, which describes alpha-mannopyranoside synthesis). On the whole, I am concerned that the introduction overstates some of the limitations of the glycal epoxide approach, and a more balanced description (along with appropriate literature citations) would be preferable.

Response:

We have included the relevant references pertaining to the reactions of 1,2-anhydrosugars and modified the discussion on the utility of this class of reagent in chemical glycosylations.

My other, and more important, concern, relates to whether this is truly a 'conceptually orthogonal approach' (page 3) relative to the methods that employ glycal epoxides. I agree that the experiments described on page 12 support a mechanism in which the free 2-OH group serves to accelerate (in the presence of a Lewis or Bronsted acid) the conversion of the anomeric stannane into a leaving group by the action of the hypervalent iodine reagent. However, this still leaves open the possibility that glycal epoxides are generated as intermediates after activation of the stannane. It is not clear why intermolecular attack by a glycosyl acceptor should compete with cyclization by the 2-OH group upon formation of an intermediate such as 59. The experiments summarized in Table 3 do not rule out the formation of a glycal epoxide from the anomeric stannane (and indeed, entries 5 and 6 confirm that such a glycal epoxide undergoes relatively efficient glycosylation under these reaction conditions).

Response:

(A) We have performed additional experiments to better understand the nature of the glycosylation intermediates. In Scheme 5 we show that (a) the ligand exchange between C2-OH and I(III) reagent results in iodine intermediate **65**, (b) both anomers of anomeric stannanes form the same semi-stable intermediate which was characterized as an anomeric tosylate (Scheme 5B).

(B) Our statement "orthogonal" refers to the fact that we use glycosyl donors with carboanionic reactivity (a nucleophile) as opposed to previous approaches that use glycosyl donors with an electronegative element at C1 (an electrophile).

The fact that the stannane-based couplings show such high 1,2-trans stereoselectivities is used as indirect support for a pathway that does not involve glycal epoxides, but this is a dangerous line of logic because data for the corresponding couplings of glycal epoxides under identical conditions do not exist. Not only are the Lewis acid catalysts not the same (e.g., gold(I) triflate in ref. 20, scandium(III) triflate in an alpha-selective example from the Gin 1998 article cited above), but the present system includes a triorganotin byproduct that may play a role by generating a stannyl ether, as suggested by the authors on page 14. In fact, couplings of stannyl ethers with glycal epoxides have been reported (e.g., Randolph and Danishefsky, J. Am. Chem. Soc. 1995, 117, 5693-5700), and provide a significant improvement over the reactions with the corresponding alcohols when hindered acceptors are used. This work should also have been cited.

A fair comparison of the present work with glycal epoxide couplings would thus require the use of acceptor-derived stannyl ethers as nucleophiles in the presence of Zn(OTf)₂. Overall, the data provided to this point do not allow for firm conclusions to be made as to whether this is truly a unique approach towards glycosylation or simply a new way to generate glycal epoxides, potentially with concurrent glycosyl acceptor activation via stannyl ether formation.

Response:

As we indicated in the first version of the manuscript, the presence of an epoxide cannot be completely ruled out. However, the accumulate evidence point to the fact that an epoxide is not the major reactive intermediate.

We also compared the reactivity of MeOSnBu₃ with MeOH in the reaction with epoxide **62** under identical conditions. Although both reactions give the expected Me-glycoside in excellent *dr*, the reaction with an epoxide is significantly slower. The reaction of an epoxide with alcohol is slower than the reaction of D-gluconal **64** with I(III) that forms formate **63**. The following points support our hypothesis that an epoxide is not the major reactive intermediate.

- (A) The rate of reaction of an epoxide with tin stannane is slower than the corresponding reaction of stannanes, Zn(OTf)₂, Koser's reagent with MeOH and MeOSnBu₃.
- (B) We reasoned that the epoxide cannot be the major reactive intermediate given that <5% of the formate product was observed under the standardized conditions for all entries in Scheme 3.
- (C) Assuming the epoxide is less reactive than the anomeric tosylate, it would be observable by NMR; however, we only observed the anomeric tosylate (Scheme 5B).
- (D) Assuming the epoxide is more reactive than the anomeric tosylate, the reaction would proceed faster when the epoxide is used; however, this is not the case.
- (E) We performed additional NMR experiments and we characterized anomeric tosylate as a stable intermediate in the reaction mixture (see scheme 5B).

Additionally, the cited paper by Danishefsky is referenced in the updated manuscript (ref. 21).

This is a pertinent issue because the stannane donors are synthesized from the glycal epoxides, and so it would be important to know whether a direct coupling of the glycal epoxides with stannylated acceptors would suffice in some instances. But it is not to say that the chemistry described here is not a useful and interesting way of generating glycosidic linkages. From my perspective, it would be preferable to focus on the potential advantages of this method in preparative carbohydrate chemistry, and to highlight the new aspects of reactivity (namely, the OH-accelerated substitution of the anomeric centre) without attempting to describe this work as being fundamentally different from glycal epoxide activation. If the authors wish to take a position on this more fundamental difference, then more experimental work aimed at ruling out glycal epoxide intermediates would be needed.

Response:

- (A) Only selected stannanes were derived from an epoxide. For example, α -stannanes and GlcNAc stannanes were obtained from anomeric chlorides
- (B) As we demonstrated in a reaction with MeOSnBu₃, even with small nucleophiles this reaction is less efficient (see p. 12, mechanistic discussion, points 3 and 4). The paper by Danishefsky mentioned above (*J. Am. Chem. Soc.* **1995**, *117*, 5693) reports only 46% of the β -anomer using a stannyl ether as a glycosyl donor.
- (C) We added additional work to demonstrate the practicality and utility of our method: (a) Scheme 3 contains four more examples of various protective groups and (b) Scheme 4 shows the synthesis of a branched oligosaccharide (**56**).

Minor points:

1. What evidence do the authors have to support the assignment of the major conformer of products **39** and **40**? The destabilizing 1,3-diaxial interaction between oxygenated substituents should more than overcome any favorable anomeric effect that might favor the 4C1 conformation for this type of α -arabinopyranoside. Assignments of signals in the ¹H NMR spectra of these compounds were not provided, so it was not clear that the major conformer was assigned based on coupling constant analysis. Incidentally, the structures of these compounds in the SI (1,2-cis) are not consistent with those shown in the paper (1,2-trans).

Response:

Regarding compounds **43** and **44** (**39** and **40** in the original manuscript), both have an anomeric peak in ¹H NMR with a coupling constant ~ 7 Hz, which indicate a diaxial ³J_{HH} coupling. Hence, we have corrected the structure drawn in Scheme 3 accordingly. For all compounds, the diagnostic C1-H was assigned in the experimental section.

2. It would be preferable to restrict the Figure captions to a simple description of what is depicted, rather than including

interpretations of the results (e.g., 'great yields and remarkably high stereospecificities', Figure 1 caption) or generalizations that merit more detailed discussion (e.g., 'When .. ROH is more hindered than a primary alcohol, a mixture of anomers is obtained', Figure 1 caption).

Response:

We modified the caption for Fig. 1 accordingly.

Reviewer #2 (Remarks to the Author):

Recommendation: The manuscript does not meet the requirements of urgency and novelty that justify publication in Nature Commun.

In the submitted manuscript Walczak and colleagues report conditions for doing efficient stereoselective oxidative glycosylation of anomeric nucleophiles with alcohols and carboxylic acids. After careful evaluation of manuscript, I do not find the manuscript suitable for publication in Nature Commun, due to the following reasons.

1. There are relatively many reports of this type transition metal catalyzed O-glycosylation (specially with the metal triflates) in the literature but the logic of exploring these tin-derivatives is weak. So I haven't found enough significance from the chemistry outlook.

Response:

We are unaware of any published work that describes O-glycosylation with carbohydrates bearing a metal at C1. Our work presents, to the best of our knowledge, the first example of chemical glycosylation between two nucleophiles. Indeed, there are many applications of metal triflates in the formation of O-glycosides but all of them involve O-, N-, S-, C-, halogen-substituted donors. The mechanism of this reaction is also distinct from the reactions described in our earlier work. We demonstrate that the unique chemical reactivity of anomeric nucleophiles such as reactions at room temperature, exclusive selectivity across a broad range of substrates, and generalized set of conditions provide a significant improvement over other methods. In our work, we use stable anomeric stannanes bearing free hydroxyl groups as the glycosyl donor, the hypervalent iodine reagents as the oxidant, and the reaction runs at room temperature. This Lewis-acid catalyzed protocol represents a practical methodology to access organic molecules containing 1,2-*trans* O-glycoside units.

2. The reaction between tin-glycoside with alcohols and carboxylic acids is bit predictable without doubt. In fact, The authors have also done similar kind of work before (J. Am. Chem. Soc., 2016, 138 (37), 12049–12052; J. Am. Chem. Soc., 2017, 139, 17908–17922). The authors haven't justified enough the novelty of their work particularly the difference from the previous reports.

Response:

Extensive literature in general organic and organometallic chemistry provides only very few parallels between C-C and C-O bond forming processes, particularly when the reactions involve C(sp³) centers. In brief, TM-catalyzed reactions proceed via different intermediates than C-O processes, thus have different stereochemical outcomes governed by numerous factors. A direct extension of a C-C bond-forming transformation into a C-O manifold is rarely a direct process and is often met with different challenges. The submitted paper describes a process that is fundamentally different than a C-C bond formation because:

- It is the first example of O-glycosylation with anomeric nucleophiles.
- Unlike C-C bond forming processes, this reaction is not stereoretentive but exclusively stereoselective. This work uses oxidants, different coupling partners and catalysts, and provides a very different stereochemical outcome.
- It is the first example of stereoselective C-O bond forming reaction from any C(sp³)-stannane (that is, outside of carbohydrate chemistry).
- The C-glycosylation of anomeric stannanes with aryl halides is redox neutral while the O-glycosylation reaction of anomeric stannanes and alcohols/carboxylic acids requires an oxidant.
- As mentioned above, our work exploits the catalytic effects of metal triflates rather than the oxidative addition/reductive elimination cycle of palladium chemistry.

3. What authors claimed in the text is the involvement of 2-hydroxyl group with hypervalent iodine, in tin-glycoside during glycosylations. It is eye-catching, but the mechanistic study for the reaction is not sufficient enough that could demonstrate the mechanistic pathway.

Response:

We have demonstrated that the C2-OH group is necessary for the reaction to take place in three different instances:

(A) 2-OBn reagents do NOT undergo O-glycosylation with iodine(III) reagents (Scheme 4). This was demonstrated in the optimization studies in Table 2, entry 10, Scheme 4A, and in the anomeric acylation studies (Table 1). We capitalized on this property to perform sequential glycosylation reactions (in the same pot) using two stannanes with and without C2-OH protection resulting in the formation of only ONE regioisomer.

(B) 2-Deoxysugars (compounds that lack C2 oxygen) do NOT undergo glycosylation with PhI(OH)OTs and other derivatives of this compound. 2-Deoxysugars are stable for at least 24 h at room temperature under these conditions without noticeable decomposition.

(C) We performed MS studies using a glycosyl donor with free C2-OH (compound 10) and Koser's reagent (PhI(OH)OTs) and we identified a ligand-exchange compound that originated from a reaction of C2-OH with Koser's reagent (m/z 945.2). Based on the identified molecular peak, we assigned the structure to the product of the reaction of C2-OH with Koser's reagent.

Taken together, we believe that the direct evidence presented above strongly supports the proposal that C2-OH acts as a directing group. Furthermore, the fact that the directing groups at C2 and the acceptor do not undergo oxidation to a ketone or aldehyde, makes this transformation particularly appealing.

Thus, I find this too weak for publication in such a high impact journal like Nature Commun. In my opinion, manuscript would be more suitable for a more specialized journal or a general chemistry journal with slightly less stringent requirements than Nature and I believe it will be more suitable in journals like ACS Catalysis or Chemical Science.

Reviewer #3 (Remarks to the Author):

Professor Walczak and coworkers have shown here an exciting work on chemical glycosylation by using glycosyl stannane donor and its new role in oxidative O-glycosylation. The manuscript is well-written and easy to understand. The supporting information such as NMR spectra are high quality, well-prepared and presented. My comments on this manuscript are;

1. The authors state that “A conceptually orthogonal approach to addresses the problem of poor anomeric stereoselectivities relies on the use of glycosyl donors with free hydroxyl groups that can act as directing groups, which will eliminate the need for the protection/deprotection steps. Herein, we report what is to the best of our knowledge, the first one-pot O-glycosylation starting from partially protected glycosyl donors under mild reaction conditions without compromising the yields and with complete anomeric selectivities” However, there are previous reports on the stereoselective glycosylation using partially protected glycosyl donors such as glycosyl dithiocarbamates by Wei and coworkers or glycosyl phosphates by Seeberger and coworkers. Please check these references;

1. Wei *et al.* *Org. Lett.* 14, 13, 3380-3383
2. Wei *et al.* *J. Org. Chem.* 79, 6, 2611-2624
3. Seeberger *et al.* *J. Am. Chem. Soc.*, 123, 9545–9554

Response:

Thank you for your positive comments and for bringing these references to our attention.

- (A) The abovementioned references have been included in the revised manuscript.
- (B) We have also modified the original sentence to more accurately present our claims and the new sentence reads as follows: “the first one-pot oxidative O-glycosylation starting from partially protected glycosyl donors under mild reaction conditions without compromising the yields and with exclusive anomeric selectivities.” We believe the fact that the reaction is an oxidative process is particularly interesting as both the 2-OH stannane and the hydroxyl acceptors may (hypothetically) undergo oxidation to a ketone or an aldehyde (not observed under our conditions).

2. A versatile glycosylation should allow chain propagation from both non-reducing and reducing end of carbohydrate. However, the authors have shown only one example of trisaccharide synthesis which is propagated from the non-reducing end.

Response:

To investigate the chain propagation from the reducing end, a protecting group on C2 of the glycosyl acceptor is needed since our method requires a free hydroxyl group on C2 to allow for chain propagation on the reducing end. In this context, we prepared 2-NAP (naphthylmethyl ether) protected stannanes with free hydroxyl groups on C6 as the glycosyl acceptors (Scheme below) and tested its reactivity. We found that only this substrate undergoes glycosylation with the C6 position in 18% yield and exclusive selectivity (the remainder of material is unreacted stannane). The deprotection of naphthylmethyl group in the presence of tributylstannane has been achieved with DDQ (82%). Additional work on the reaction conditions would be necessary to fully optimize the yield. However, we were able to demonstrate the utility of this method in extending the chain from the reducing end.

3. The scope of O-glycosylation is limited to glycosyl donors and acceptors equipped with benzyl ether or ester group. Thus, I wonder about its application toward those less stable protecting groups which are necessary for the orthogonal protecting group manipulations for complex carbohydrates.

Response:

We have added four examples in the revised manuscript with 2-naphthylmethyl (Nap), *p*-methoxybenzyl (PMB), levulinyl (Lev), and diacetonide (acetal) used as orthogonal protecting groups. These groups represent a selection of the most common groups in the current preparative carbohydrate chemistry, are compatible with the current reaction conditions, and provide the corresponding products with exclusive anomeric selectivities. Scheme 3 was updated accordingly (compounds **34**, **35**, **37**, and **39**).

4. Authors should provide a synthesis of more complex oligosaccharide to emphasize practicality of this oxidative glycosylation.

Response:

We added a new example of sequential glycosylation forming branched oligosaccharide **56** (Scheme 4B). In addition to orthogonal reactions from Scheme 4A and the synthesis of linear oligosaccharides from Scheme 4C, this reaction sequence forms a structurally complex and congested glycan with a 1,4-linkage. The key practical elements of our methods are:

(A) the use of standardized conditions where the C2-OH group directs regio- and chemoselectivity (other methods often require two different classes of leaving groups; in our case, the same leaving group is tolerated),

(B) the reaction does not require cooling, can be performed at rt, and is suitable for automation,

(C) reagents and substrates are indefinitely stable.

5. For moderate yields on the glycosylation in Scheme 3, could authors give comment on the byproduct?

Response:

We observed three decomposition pathways of the starting material stannane in the oxidative glycosylation:

(A) The formation of glycals when a Lewis acid is used, presumably through the activation of the hydroxyl group, followed by an elimination. This pathway takes place when the glycosyl acceptor is absent.

(B) The cleavage of C1-C2 in the stannane to form formate **62** (numbering in the revised manuscript).

(C) C2-C3 olefinated glycoside has been isolated in several cases, supposedly from the isomerization of glucal catalyzed by the Lewis acid.

Comments specific to reactions in Scheme 3 are included in the updated manuscript.

6. The use of glycosyl stannane itself may be challenging to others especially those who are not in a field of organometallic chemistry. Could authors give comment on the removal of organotin after the glycosylation.

Response:

Tributyltin by-products are easily removed by silica gel column chromatography due to the significant difference in polarity between stannanes and glycosides. Hexanes and toluene are excellent solvents for this purpose. Also, saturated aqueous KF wash prior to the column chromatography is helpful. We added a comment in the manuscript about the purification and removal of tin by products.

7. Please check the consistency of reference format.

Response:

The reference formatting has been updated.

In summary, I will accept this manuscript if the authors could provide more examples to address on my comment number 3, and 4.

Reviewer #1 (Remarks to the Author):

The revised manuscript represents an improvement on the original in several respects, and in particular the new mechanistic hypothesis involving an anomeric sulfonate appears to be much more reasonable than what had been proposed initially. Considering the new mechanistic insight, the continued characterization of this process as being 'conceptually orthogonal' seems increasingly difficult to justify. This is ultimately a normal-polarity glycosylation involving a nucleophilic substitution at the anomeric center. In any case, the mode of reactivity and the unique activating effect of the 2-OH group may be of interest to readers of Nature Communications, and I recommend that the manuscript be accepted for publication.

Minor points:

1. The discussion of approaches to stereocontrolled glycosylation should include references to catalyst-controlled methods (e.g., Jacobsen and co-workers, Science 2017, 355, 162-166).
1. It would be appropriate to cite Bohé and Crich's review on glycosyl sulfonates (Chapter 6 in Selective Glycosylations, C. S. Bennett, Ed.).

Reviewer #3 (Remarks to the Author):

The revised manuscript from Professor Walczak and coworkers has addressed my comments. The authors expand scope of the acceptor to other protecting groups and also demonstrated more examples of oligosaccharide synthesis (however, I would prefer to see more complex structure but it's understandable due to time constraint). Merit of this work based on glycosylation between nucleophilic donor and nucleophilic acceptor. To the best of my knowledge, it is unprecedented before. This could initiate a new direction for chemical glycosylation study. Thus, I recommend this manuscript to publish in Nature Commun.

Minor points; please check several typos as follows;

Page 8 line 4; ... catalyst loading without.

Page 9 line 4; ... resulted in good to excellent yields... However, the product 35 was obtained in only 53% (good?).

Page 11 line 4; ... the pivaloyl eester group.

Page 12 line 9-11; ... "Schmidt reported that 2-hydroxy-1-acetimidate, upon activation with a Brønsted acid, undergoes a glycosylation resulting in a mixture of anomers. Similarly, Baker found

that 2hydroxy-1-thioglycosides, in a reaction with a Lewis acid, resulted in a mixture of anomers.”
What is the authors trying to explain?

RESPONSE TO REVIEWERS

We would like to thank the reviewers for their time and effort reviewing the revised manuscript. The following document contains a response to their comments.

REVIEWERS' COMMENTS:

Reviewer #1 (Remarks to the Author):

The revised manuscript represents an improvement on the original in several respects, and in particular the new mechanistic hypothesis involving an anomeric sulfonate appears to be much more reasonable than what had been proposed initially. Considering the new mechanistic insight, the continued characterization of this process as being 'conceptually orthogonal' seems increasingly difficult to justify. This is ultimately a normal-polarity glycosylation involving a nucleophilic substitution at the anomeric center. In any case, the mode of reactivity and the unique activating effect of the 2-OH group may be of interest to readers of Nature Communications, and I recommend that the manuscript be accepted for publication.

Minor points:

1. The discussion of approaches to stereocontrolled glycosylation should include references to catalyst-controlled methods (e.g., Jacobsen and co-workers, Science 2017, 355, 162-166).

1. It would be appropriate to cite Bohé and Crich's review on glycosyl sulfonates (Chapter 6 in Selective Glycosylations, C. S. Bennett, Ed.).

Response: These two papers are cited in the updated manuscript.

Reviewer #3 (Remarks to the Author):

The revised manuscript from Professor Walczak and coworkers has addressed my comments. The authors expand scope of the acceptor to other protecting groups and also demonstrated more examples of oligosaccharide synthesis (however, I would prefer to see more complex structure but it's understandable due to time constraint). Merit of this work based on glycosylation between nucleophilic donor and nucleophilic acceptor. To the best of my knowledge, it is unprecedented before. This could initiate a new direction for chemical glycosylation study. Thus, I recommend this manuscript to publish in NatureCommun.

Minor points; please check several typos as follows;

Page 8 line 4; ... catalyst loading without.

Page 9 line 4; ... resulted in good to excellent yields... However, the product 35 was obtained in only 53% (good?).

Page 11 line 4; ... the pivaloyl eester group.

Response: The following typos were corrected in the updated manuscript.

Page 12 line 9-11; ... "Schmidt reported that 2-hydroxy-1-acetimidate, upon activation with a Brønsted acid, undergoes a glycosylation resulting in a mixture of anomers. Similarly, Baker found that 2hydroxy-1-thioglycosides, in a reaction with a Lewis acid, resulted in a mixture of anomers." What is the authors trying to explain?

Response: We reference prior work to indicate that under classical activation conditions that may result in the formation of an epoxide, a mixture of anomers is formed. This indirect evidence supports our claim that an epoxide is not likely to be a reactive intermediate under our conditions.